# Learning to Control Latent Representations for Few-Shot Learning of Named Entities

## Abstract

Humans excel in continuously learning with small data without forgetting how to solve old problems. However, neural networks require large datasets to compute latent representations across different tasks while minimizing a loss function. For example, a natural language understanding (NLU) system will often deal with emerging entities during its deployment as interactions with users in realistic scenarios will generate new and infrequent names, events, and locations. Here, we address this scenario by introducing a RL trainable controller that disentangles the representation learning of a neural encoder from its memory management role.

Our proposed solution is straightforward and simple: we train a controller to execute an optimal sequence of read and write operations on an external memory with the goal of leveraging diverse activations from the past and provide accurate predictions. Our approach is named Learning to Control (LTC) and allows few-shot learning with two degrees of memory plasticity. We experimentally show that our system obtains accurate results for few-shot learning of entity recognition in the Stanford Task-Oriented Dialogue dataset.

## 1 Motivation

Today, supervised models have problems incorporating new tasks over time while protecting previously acquired knowledge. This is because these algorithms require that all data is given prior to training. This becomes a problem in the presence of more general scenarios in which new classes emerge during training or data exhibit long-tailed distributions. Hence, classes with large support dominate the learning of gradient-based representations causing the *catastrophic forgetting* of under-represented classes, Kirkpatrick et al. (2016).

Unlike deep neural networks, humans and other mammals exel in learn incrementally with small data. Biological evidence suggests that the process of acquiring new skills occurs in different brain areas with at least two degrees of plasticity, Wixted et al. (2018): 1) A *slow* memory that is dense and requires extensive practive. 2) A *fast* memory that is sparse and stores volatile information. Inspired by these observations, we introduce a trainable controller, Learning to Control (LTC), that learns to interact with both a *slow* memory (consisting of neural encoders) and a *fast* memory (consisting of an associative array storing key-value pairs). We experimentally show the advantage of using LTC for a few-shot learning setup. Figure 1 depicts the network architecture of our model.

We present the following contributions:

- We propose a novel architecture that uses a trainale controller to manipulate latent representations in an external memory (Section 3).

- We introduce the use of a reward signal that is proportional to the average reduction of entropy when attending the memory entries. This enables us to propose a reinforcement learning approach that learns a policy based on interactions with the external memory.

- We show the generality of our solution for the few-shot learning of entities in the Stanford Task-Oriented Dialogue dataset.

## 2 DENSE AND SPARSE MEMORIES

The proposed architecture considers the use of two types of memories that behave differently during backpropagation. First, we define a memory as follows.

**Definition 2.1. (Memory)** Given a neural model $f(x, y, \theta)$ that maps an observation $x$ to the learning task $y$ and is parameterized by $\theta$. Then, the memory of $f(x, y, \theta)$ is a collection of co-activations responding similarly to the reocurring patterns associated to the learning task $y$.

A Neural Network uses most of its memory to remember patterns in the dataset. However, our approach distinguishes between *dense* and *sparse* memories.

**Definition 2.2. (Dense Memory)** A dense memory is a type of memory, in which all its learnable parameters $\theta$ have the possibility of being updated during training.

We want to preserve latent representations of the input between training steps but a dense memory will likely overwrite them based on frequent global updates. We then need a sparse memory with addressable memory entries and sparse updates.

**Definition 2.3. (Sparse Memory)** Given a model $f(x, y, \theta)$, let $a(x, y)$ be a memory operation which returns $k$ updating parameters to incorporate the pair $(x, y)$. Then, a sparse memory is a type of memory that satisfies the *sparse updating constraint*:

$$0 < |a(x, y)| \ll |\theta|$$

where $|\theta|$ is the total number of trainable parameters.

We use the sparse memory introduced in Kaiser et al. (2017) which consists of two arrays of size `mem_size` for storing keys $(K)$ and values $(V)$. Each key has a dimensionality of `key_dim` and each value is a scalar representing a class label. In our experiments, we fixed the number of updating parameters, i.e. keys, to be very small $(k = 10)$ in comparison to `mem_size` in order to satisfy the *sparse updating constraint*, $k \ll$ `mem_size`. The final form of the sparse memory is as follows.

$$M := (K_{mem\_size \times key\_dim}, V_{mem\_size})$$

The state of a sparse memory changes based on the following memory operations.

**Definition 2.4. (Memory Operations)** Let $s$ be a input embedding that we want to incorporate into $M$, $y$ be a class label, and $i$ be the memory index of the most similar key to $s$ based on the Cosine similarity.

1. If $V[i] \neq y$, the $write(i, s, y)$ operation registers a new class with $M[i] = \langle s, y \rangle$.

2. If $V[i] = y$, the $update(i, s, y)$ operation modifies the stored key-based representation $K[i]$ with $M[i] = \langle \|K[i] + s\|, y \rangle$.

## 3 LEARNING TO CONTROL

The goal of Learning To Control (LTC) is to learn a strategy to manage the entries of a sparse memory. We characterize this decision process with a memory policy $\pi_\theta$ which represents the probability of running an memory operation given the current state of the memory. Rathern than following a heuristics (e.g., overwritting in the oldest memory entry as in Kaiser et al. (2017)), learning such policy has the advantage of optimizing the allocation of infrequent classes considering the limited number of memory entries in $M$. Indeed, $\pi_\theta$ transitions $M$ to a new configuration in which its keys are more adapted to store and maintain informative latent vectors. To clarify the connection between the components, let us consider the forward and backward computations involved in LTC.

### 3.1 FORWARD COMPUTATION

As illustrated in Figure 1, information flows through the hidden layers of the neural encoder, feeds the controller, and reaches the external memory $M$. The encoder is a Long-Short Term Memory (LSTM)

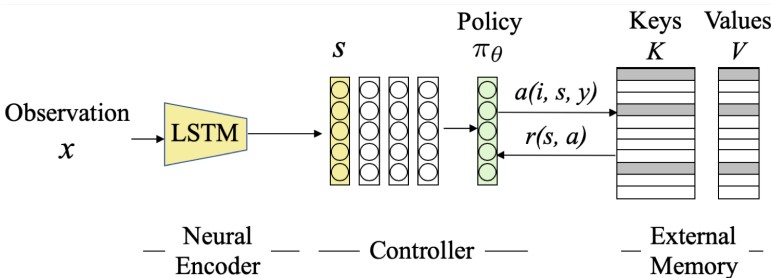

Figure 1: Architecture of our approach based on a neural encoder that generates input representations and a trainable controller that learns to store and maintain them in an external memory formed by key-value pairs. The memory is modified via operations $a$, each of which returns a reward signal $r$.

neural network that transforms the raw input $x$ into the last hidden state of the following recurrence $h_i = LSTM(x, h_{i-1})$, where $s = h_i$ is the state of the environment prior to any memory operation and corresponds to the last hidden state of the LSTM encoder.

Which memory entries should the controller update to incorporate the learning task $(s, y)$? Our approach is to induce the probablity distribution $\pi_\theta(i|s)$ over the memory entries given the state $s$ and sample from there the most likely memory index $i_{\pi_\theta}$

$$\pi_\theta(i|s) = Softmax(s \cdot K_i)$$
$$i_{\pi_\theta} \sim argmax_i(\pi_\theta(i|s)).$$

Then, the controller executes the memory operation $a(i_{\pi_\theta}, s, y)$ at position $i_{\pi_\theta}$ considering the Definition 2.4. During inference, $\hat{y} = V[i_{\pi_\theta}]$ returns the predicted class of the raw input $x$ .

## 3.2 BACKWARD COMPUTATION

The forward step performs an operation $a(i_{\pi_\theta}, s, y)$ in the sparse memory and modifies its keys to integrate latent representations. The backward step estimates the objective function $J(\theta)$ and computes its gradients with respect to the trainable parameters of the encoder and controller modules.

In the case of the controller, we want to compute the gradient vector $\nabla_\theta J(\theta)$ that updates the stochastic policy $\pi_\theta$ in a direction that maximizes $J(\theta)$ during few-shot learning. For our purposes, each memory operation $a$ returns a reward $r$ that measures the efficacy of this operation to reduce uncertainty about which of the keys in $K$ will lead the transition between memory states. This means going from a state in which no operation took place in memory ($K_{t-1}$) to an updated state ($K_t$). More formaly, our reward function $r_t(s, a)$ corresponds to the reduction of entropy $H$ when softly attending over the memory entries before and after executing the operation $a$.

$$h_{K_t} = s \cdot K_t$$
$$att_{K_t} = Softmax(h_{K_t})$$

We use the dot-product attention score Luong, Pham, and Manning (2015) to compute the logits $h_{K_t}$ over the memory keys and use Softmax to generate the probability distribution $att_{K_t}$. Zhu et al. (2005) demonstrates that models that show high entropy in their output distributions often fail to discriminate classes because of exhibiting uniform-like distributions and a high degree of uncertainty. Thus, we use the following entropy-based reward signal to encourage the use of few memory entries to modify the sparse memory $M$.

$$r_t = -\big(H(att_{K_t}) - H(att_{K_{t-1}})\big)$$

Based on memory operations, we can approximate the objective function from a batch of sampled episodes of length $T$ as the expected value of the cumulative sum of rewards under the policy $\pi_\theta$,

$$J(\theta) = \mathbb{E}_{\pi_\theta}\left[\sum_{t=1}^{T} r_t(s, a)\right].$$

| Model | 5-way 1-shot | 5-way 5-shot | 20-way 1-shot | 20-way 5-shot |
|---|---|---|---|---|
| Memory Augmented NN | 63.1% | 66.7% | 57.2% | 61.3% |
| Matching Networks | 67.3% | 71.4% | 62.2% | 68.3% |
| Learning to Control | **71.5%** | **77.1%** | **65.3%** | **75.8%** |

Table 1: Average accuracy for few-shot learning in the STDO dataset.

The REINFORCE algorithm Williams (1992) allow us to directly take the gradients of the objective function $J(\theta)$ as described in Equation 1 and used them to perform backpropagation.

$$\nabla_\theta J(\theta) \approx \frac{1}{N} \sum_{n=1}^{N} \left( \sum_{t=1}^{T} \nabla_\theta \log \pi_\theta(i_n^t | s_n^t) \right) \left( \sum_{t'=t}^{T} r_{t'}(s_n^{t'}, a_n^{t'}) \right) \tag{1}$$

## 4 EXPERIMENTS

We evaluate our proposed method on the Stanford Task-Oriented Dialogue Dataset (STDO) Eric et al. (2017) which consists of $3,031$ dialogues in the domain of an in-car assistant that provides automatic responses to the requests of a driver considering weather, point-of-interest, and scheduled domains.

### 4.1 BASELINES

We compare Learning to Control (LTC) with the following baselines: 1) **Matching Networks** (MN), Vinyals et al. (2016): This algorithm provides one-shot learning capabilities by jointly mapping a small labeled support set and an unlabeled example to its most likely label, and 2) **Memory Augmented Neural Networks** (MANN), Santoro et al. (2016): This algorithm is designed to provide one-shot learning based on a Neural Turing Machine and a curriculum training regime.

### 4.2 FEW-SHOT LEARNING FOR ENTITY RECOGNITION

We study the problem of sequence labeling to find the best label sequence (named entities) for a given input sentence considering an LSTM encoder augmented with an external memory to store hidden representations for each recurrent unit. We use the Stanford Named Entity Recognizer (NER)[1] to augment the STDO dataset with 7 classes (Location, Person, Organization, Money, Percent, Date, Time) and an non-entity class. This results in $15,928$ observations, 8 classes, and an average sequence length of 44 words.

We change the format of the STOD dataset to simulate a scenario in which classes obtain incremental support during training. This is done by forming episodes of labeled examples that show a uniform distribution over $k$ classes (e.g., 5-way), so we can track the number of times a particular class was presented to the model during training (e.g., 1-shot learning). This format was proposed by Santoro et al. (2016) to study *few-shot learning*. The goal of this experiment is to learn with small data, so we study the learning behavior of LTC with an external memory of $1,000$ entries and in relation to two models designed to provide few-shot learning capabilities: Memory Augmented Neural Networks, Santoro et al. (2016) and Matching Networks, Vinyals et al. (2016).

Table 1 shows that LTC provides an average improvement of $+5.2$ in the STOD dataset with respect to the MN neural model, which is consistently the second most accurate option for both 5-way and 20-way classification and when the number of classes ranges from small (5-way) to large (20-way). Similarly, LTC also shows an average improvement of $+10.35\%$ with respect to the MANN model. For the scenario of learning with infrequent classes, LTC outperforms MN when only 1 training observation is presented (1-shot) and 5 and 20 classes are available during training by $+4.2\%$ and $+3.1\%$, respectively. The performance advantage in the results can be explained by the extra memory capacity of LTC and represent an opportunity to explore later what is the minimum memory size to continuously learn with small data, an even more challenging problem to study in the future.

---

[1]`https://nlp.stanford.edu/software/CRF-NER.html`

## 5    RELATED WORK

The idea of training a controller that interacts with a memory to incrementally allocate distributional representations builds upon a wide range of research in machine learning and memory augmented algorithms.

Kraska et al. (2018) introduced the concept of learned indices structures as a learning architecture that predicts the position or existence of records. Their main goal is to prevent too many collisions from being mapped to the same position inside the hash index. This is because collisions often cause memory overhead when traversing a linked list or require the allocation of additional memory for storing more records. They approach this problem as supervised learning of the cumulative distribution function of hash keys, which leads to minimizing the number of collisions. In contrast, we propose the use of an associative array as an indexed structure that supports generalization, so the collision of multiple observations to a similar hash key if adequate for predicting the class of similar input data. While the idea of learning hash functions as neural networks is not new, existing work mainly focused on learning a better hash-function to map observations into low-dimensional embeddings for similarity search assuming a fixed data distribution Qian et al. (2014). To our knowledge, it has not been explored yet if it is possible to learn a hash index according to a data distribution in which few training observations are presented per class during training.

Deep Neural Networks are models that solve classification problems with non-linearly separable classes. These are shown to be good for problems such as object detection Krizhevsky, Sutskever, and Hinton (2017), language translation Johnson et al. (2016) and image generation Goodfellow et al. (2014). Recently, recurrent neural networks (RNN) have been studied to manage the states between time steps using an attention mechanism that addresses similar content Bahdanau, Cho, and Bengio (2014). Some similar approaches have also studied the problem of few-shot learning Santoro et al. (2016)Koch, Zemel, and Salakhutdinov (2015)Vinyals et al. (2016), a representative example is Matching Networks Vinyals et al. (2016), which uses an attention mechanism to augment neural networks for set-to-set learning.

A relevant research topic to this work is the idea of memory augmented networks that extend its capabilities by coupling an external memory. For example, Kaiser et al. (2017) also proposes a key-value memory, but with no controller mechanism for adequate training of few-shot learning tasks. Also, Neural Turing Machines (NTM) Graves, Wayne, and Danihelka (2014) are differentiable architectures allowing efficient training with gradient descend and showing important properties for associative recall for learning different sequential patterns. Although they have important properties for one-shot learning given their sequential memory management, the supervised nature of its training still shows issues related to catastrophic forgetting.

## 6    CONCLUSIONS

Learning new tasks without forgetting the previously ones references the plasticity of our own brain to retain previously acquired knowledge. In this work, we show a learning system that stores information about named entities with two levels of representation: dense and sparse. While a dense memory captures features that are shared across tasks, sparse memory incorporates embedding vectors and prevents them from being overwritten by global updates during backpropagation. To do this, we rely on the sparse structure of associative arrays to locally update a small set of memory entries and on the use of a trainable controller that manages the transition of the sparse memory to a state of low entropy that also addresses the learning task. This enables a model to incrementally learn in the presence of a few observations per class. We experimentally show that our system obtains accurate results for few-shot learning of entities in the Stanford Task-Oriented Dialogue dataset in comparison with state-of-the-art few shot learning algorithms (Memory Augmented Neural Networks and Matching Networks).

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
