# OpenReview forum: "Learning to Control Latent Representations for Few-Shot Learning of Named Entities"
_ICLR.cc/2020/Conference — Reject_

### Official Review · AnonReviewer3 · 2019-10-22
**Official Blind Review #3**

**Rating:** 3

**Review:**

In this paper, the authors present a method, Learning to Control (LTC), that enables a reinforcement learning agent to learn to read and write external memory. They follow the intuition that human has two degrees of plasticity for memory, which leads to the dense-sparse memory design in this paper. The proposed method can be applied to a few-shot setting.

writing: The writing is ok. Readers can obtain the essence by a reasonable amount of guessing.


Strength: Although memory augmented neural networks and reinforcement learning are both well-known, the proposed method combined them in a novel way.  The presented approach outperforms two baselines on the stanford dialogue dataset.

Weakness: There is no ablation study for the proposed method to fully understand why the proposed method is superior. The authors use REINFORCE as the RL algorithm which is no longer useful in most of the complex RL tasks. More advanced RL algorithms are encouraged to be tried. The proposed title is "Learning to Control",   which I don't see where the control part is. It seems the LTC only refers to the REINFORCE agent. The experiments are very minimal which did not provide enough information to distinguish the proposed method from other methods.

Overall, the general direction is promising. However, the paper is not yet fully finished. I believe the authors need major changes to the experiments section as well as the method description section. Hence, I cannot recommend this paper to acceptance.

minor:
exel -> excel
Figure 1 needs more clarification.

**Experience Assessment:**

I have read many papers in this area.

**Review Assessment: Checking Correctness Of Derivations And Theory:**

I assessed the sensibility of the derivations and theory.

**Review Assessment: Checking Correctness Of Experiments:**

I carefully checked the experiments.

**Review Assessment: Thoroughness In Paper Reading:**

I read the paper at least twice and used my best judgement in assessing the paper.

---

### Official Review · AnonReviewer2 · 2019-10-22
**Official Blind Review #2**

**Rating:** 1

**Review:**


The paper proposes a memory network architecture with a sparse memory. Each memory entry contains a key (vector) and a value (class label). The memory is addressed using a policy pi_theta, which selects a single memory entry to be updated at each time step. The policy pi_theta is trained using policy gradient with the reward being the increase in the policy's certainty (measured as entopy). The model is evaluated on an online NER task that mimics the meta-learning setting of http://proceedings.mlr.press/v48/santoro16.html. In that work, the examples are provided in episodes. The labels are renamed at the beginning of each episode (to prevent fitting to the labels). The model has to predict the label of each example in sequence, and the correct label is given after each prediction.

The paper suffers from the following issues:

- The method in the paper is very similar to the existing discrete addressing scheme (https://www.mitpressjournals.org/doi/full/10.1162/neco_a_01060), which naturally also uses RL for training.

- The method description is either incomplete or incorrect. The task label does not seem to influence the loss function at training time. The current training loss only encourages the entropy to reduce, so unless the LSTM is pre-trained, there is no guarantee that the key s will address the right memory entry. Please correct me if I am mistaken here.

- The writing is unclear. Task descriptions and notations are not properly set up. The experiment section does not explain the details of the experiments, leaving the reader to look at Santoro et al., 2016 for context. Even then, some statements in the experiment section are confusing. For instance, the numbers of classes in the description (5 and 20) do not match the number of NER classes (8).

- The task used in the paper is non-standard. The paper could have used meta-learning tasks from previous papers, including the ones in Santoro et al., 2016.

Other comments:

- The citations should be written as "xxx (author, year)" and not "xxx, author (year)".
- The paper could benefit from proofreading.

**Experience Assessment:**

I have read many papers in this area.

**Review Assessment: Checking Correctness Of Derivations And Theory:**

I carefully checked the derivations and theory.

**Review Assessment: Checking Correctness Of Experiments:**

I assessed the sensibility of the experiments.

**Review Assessment: Thoroughness In Paper Reading:**

I read the paper thoroughly.

---

### Official Review · AnonReviewer1 · 2019-10-27
**Official Blind Review #1**

**Rating:** 1

**Review:**

The authors build on work regarding few-shot learning with memory augmented networks, specifically [Kaiser, et al., ICLR17] where the goal is to learn a memory address mapping such that generalization is achieved by finding the nearest neighbor memory address when predicting the label. For correct predictions, the memory key is updated to include the associated (predicted) address while new memory locations are written for mistakes. Whereas [Kaiser, et al., ICLR17] follows a LRU-like procedure for replacing memory, the current work proposes performing policy-gradient RL where the action space is the memory locations and the reward is reduction of entropy over the memory address assignment distribution over the memory locations. This approach is empirically studied for an RNN approach to NER, specifically considering few-shot learning for NER in the Stanford Task-Oriented Dialogue (STOD) dataset — showing non-negligible improvements over Memory Augmented Networks [Santoro, et al., ICML16] and Matching Networks [Vinyals, et al., NeurIPS16].

From a high-level perspective, memory networks have played an important role in building dialogue managers (and in several applications) and their proposed expansion over [Kaiser, et al., ICLR17] in the context of few-shot, sparse memory architectures seems sensible. Additionally, the empirical results o NER seem promising. However, in my opinion, this paper is making contributions along two dimensions, but neither of them convincingly as detailed below.

First, one potential point of emphasis is the ‘learning to control’ mechanism through PG RL being better than [Kaiser, et al., ICLR17] using with a LRU-like policy. To do this, it would make sense to both compare to the datasets in the previous work (Omniglot, WMT — even if it does require additional architectures that are pretty straightforward expansions) and to include this architecture in the submitted paper as there is a publicly available dataset (https://github.com/himani-arora/learning_to_remember_rare_events). While the synthetic dataset probably isn’t necessary, one could easily think of synthetic datasets that would clearly contrast with the existing method and conduct ablation/configuration studies regarding different number of neighbors, size of memory, exact vs. approximate nearest neighbor for lookup, etc.

A second point of interest is the NER task. From this perspective, there is some existing work (e.g., [Fritzler, Logacheva & Kretov, SAC19; Hou, et al., https://arxiv.org/pdf/1906.08711.pdf (Not required as not accepted yet); Hofer, et al., https://arxiv.org/pdf/1811.05468.pdf (Not clear if accepted)]. While these are largely preliminary works, they do provide datasets (and there are many others) for NER that would potentially make a convincing case for this being a state-of-the-art approach for few-shot learning in NER. However, I am not sure that being the state-of-the-art for few-shot NER would be a sufficient contribution for ICLR, even if done convincingly. If there was something in the algorithm specific to sequence prediction, then I think few-shot sequence prediction would be a sufficient contribution, but given the current architecture, I would think that the bar is a methodological contribution to adding sparse memory units (via RL in this case) is the acceptable level of contribution.

Finally, my belief that the contribution is too narrow and not sufficiently developed is supported by the writing seeming rushed. The paper makes much more sense after reading [Kaiser, et al., ICLR17] as Sections 2, 3 of the submission are missing the aspects of a clear constrast to [Kaiser, et al., ICLR17], a working example to provide readers a more intuitive understanding, a clear example showing dimensionality and notation (beyond what is in Figure 1). There are also many typos (e.g., “mammals exel”, “trainale controller”, “Rathern then”, amongst others) and general need for proofreading. However, this is not the determining factor in the paper. Basically, I think the contribution is insufficient in terms of scope and convincingness of the contribution. Thus, I recommend rejecting in its current form — even if the underlying idea is promising and worth developing further.

**Experience Assessment:**

I have read many papers in this area.

**Review Assessment: Checking Correctness Of Derivations And Theory:**

N/A

**Review Assessment: Checking Correctness Of Experiments:**

I carefully checked the experiments.

**Review Assessment: Thoroughness In Paper Reading:**

I read the paper thoroughly.

---

### Decision · Program_Chairs · 2019-12-19

**Decision:**

Reject

**Comment:**

This work proposes to use policy-gradient RL to learn to read and write actions over memory locations using as reward the entropy reduction of memory location distribution. The authors perform experiments on NER in Stanford Dialogue task, that are framed though as few-shot learning. The reviewers have pointed out shortcomings of the paper with regards to its novelty, narrow contribution in combination thin experimental setup (the authors only look into one dataset and one task with minimal comparison to previous work and no ablation studies as to understand the behaviour of the model) and clarity (method description seems to be lacking some crucial components of the model). As such, I cannot recommend acceptance but I hope the authors will use the reviewers comments to transform this into a strong submission for a later conference.